# Pyroptosis and Insulin Resistance in Metabolic Organs

**DOI:** 10.3390/ijms231911638

**Published:** 2022-10-01

**Authors:** Huiting Wei, Di Cui

**Affiliations:** College of Physical Education, Hunan University, Changsha 410012, China

**Keywords:** pyroptosis, IR, skeletal muscle, NLRP3, caspase, GSDMs

## Abstract

Skeletal muscle serves as the optimal effective organ to balance glucose homeostasis, but insulin resistance (IR) in skeletal muscle breaks this balance by impeding glucose uptake and causes metabolic disorders. IR in skeletal muscle is caused by multiple factors, and it has been reported that systemic low-grade inflammation is related to skeletal muscle IR, though its molecular mechanisms need to be ulteriorly studied. Pyroptosis is a novel inflammatory-mediated type of cell death. It has recently been reported that pyroptosis is associated with a decline in insulin sensitivity in skeletal muscle. The appropriate occurrence of pyroptosis positively eliminates pathogenic factors, whereas its excessive activation may aggravate inflammatory responses and expedite disease progression. The relationship between pyroptosis and IR in skeletal muscle and its underlined mechanism need to be further illustrated. The role of pyroptosis during the process of IR alleviation induced by non-drug interventions, such as exercise, also needs to be clarified. In this paper, we review and describe the molecular mechanisms of pyroptosis and further comb the roles of its relevant key factors in skeletal muscle IR, aiming to propose a novel theoretical basis for the relationship between pyroptosis and muscle IR and provide new research targets for the improvement of IR-related diseases.

## 1. Introduction

Insulin resistance (IR) refers to a state in which the biological effect of a given insulin concentration is reduced [1]. IR is primarily caused by the impaired ability of the adipose tissue to store excess energy as fat and impaired insulin receptor signaling, occurring frequently in the liver, adipose tissue and skeletal muscle [2,3]. In recent years, IR in skeletal muscle has been recognized as the major pathological condition associated with metabolic syndrome, which subsequently develops into type 2 diabetes and accompanying complications [4]. Other studies have shown that skeletal muscle IR is related to long-term hypodense inflammation and contributes to the chronic damage and dysfunction of various tissues and organs [5,6]. Some clinical data also suggest a link between low-grade inflammation and metabolic diseases. For example, among Chinese patients with confirmed COVID-19, more than 20% of the underlying disease was found to be diabetes, and the fatality rate appeared to be as high as 7.3%, meaning that COVID-19 and diabetes may have mutually amplified positive feedback effects [7]. Pyroptosis, a newfound special programmed cell death form directly related to inflammation, exerts a protective role in the innate immune response by removing intracellular pathogens through the inflammatory reaction, which inhibits intracellular pathogen replication and activates the immune cells promoting the engulfment and killing of pathogens [8]. However, when the cells encounter a fiercer pathogen stimulation, pyroptosis can be overactivated and initiate a series of diseases [9]. Pyroptosis came into the attention of researchers during a macrophage infection experiment in 1992, in which the morphology of dying cells showed characteristic features distinguished from those of apoptosis, such as osmotic swelling and lysis, assembly of the cell membrane into pores, extracellular release of cytosolic content, and expanded inflammatory response [10,11,12,13]. Pyroptosis relies on the activated pathways associated with the NOD-like receptor thermal protein domain associated protein 3 (NLRP3) inflammasome to magnify a chronic low-grade inflammation; the cascade amplification caused by the release of cell content and proinflammatory factors may be one of the newly discovered pathological mechanisms in diabetes [14,15]. Considering the inflammatory reaction as a breakthrough in diabetes research and reviewing relevant studies, it was found that the early occurrence and subsequent development of skeletal muscle IR might be interwoven with pyroptosis. It is thus of great value to clarify the pathological mechanism underlying low-grade chronic metabolic inflammation and IR development in skeletal muscle [16]. However, the relationship between pyroptosis and skeletal muscle IR needs to be further understood. In addition, the role of pyroptosis during the process of IR alleviation induced by non-drug interventions, such as exercise, needs to be clarified. Therefore, we reviewed the main molecules and pathways of pyroptosis, elucidated the intrinsic link between pyroptosis molecules and skeletal muscle IR and propose a possible mechanism by which exercise improves skeletal muscle IR and pyroptosis, suggesting new research directions and potential targets to counteract skeletal muscle metabolic inflammation.

## 2. Pyroptosis

### 2.1. Molecular Mechanism of Pyroptosis

The inflammasome complex, a vital element in the activation of the pyroptosis pathway in response to external pathogens, is essential for the occurrence and maintenance of the inflammatory response. It is activated by pathogenic signals such as pathogen-associated molecular patterns (PAMPs) and damage-associated molecular patterns (DAMPs) and enhances local and systemic inflammatory responses [17,18,19,20]. The inflammasome complex consists of NOD-like receptors (NLRs), apoptosis-associated speck-like protein containing a CARD (ASC), pro-caspase-1, absent in melanoma 2 (AIM2) and Pyrin [21]. Related to pyroptosis in NLRs are NLRP1, NLRP3 and NLRC4, and NLRP3 is the most widely studied [22]. When internal or external stimuli are sensed, the PYD structure of NLRP3 binds to the PYD domain of ASC, and the CARD domain of ASC and the CARD domain of pro-caspase-1 also interact, so that these molecules eventually clump together to form the NLRP3 inflammasome [23]. Pro-caspase-1, the predecessor of caspase-1, is cleaved, forming a heterotetramer, which is further converted to the active form of mature caspase-1 [24]. AIM2, a receptor of cytoplasmic DNA, can activate caspase-1 by forming a complex with its ligand and ASC, and the knockout of AIM2 can eliminate caspase-1 activation by cytoplasmic dsDNA and dsDNA virus [25]. Pyrin is a cytosolic pattern recognition receptor that regulates innate immune responses when detecting PAMPs/DAMPs and assembles inflammasomes by combining with several other receptors, which recruit and activate caspase-1 [26,27].

Gasdermins (GSDMs) are a protein family consisting of six proteins encoded by paralogous genes, in addition to DFNB59, GSDMA, GSDMB, GSDMC, GSDMD and GSDME (also known as DFNA5) all performance pore-forming activity [28,29]. Among them, the main ones involved in pyroptosis are GSDMD and GSDME, which widely exist in different cell tissues [30]. GSDMD is composed of an N-terminal domain (Gasdermin D-NT, GSDMD-NT), a domain linker, and a C-terminal domain (Gasdermin D-CT, GSDMD-CT). GSDMD-NT is cytotoxic but is in an autoinhibited state when bound to the GSDMD-CT [31,32]. After GSDMD is cleaved, it releases GSDMD-NT, which will assemble into pores in the cell membrane. These will continuously expand, forming huge bubbles and releasing the cell content, including inflammatory cytokines, so that the dying cell becomes flattened [33]. The loss of GSDMD delays cell membrane rupture and changes the type of death from pyroptosis to apoptosis [34]. GSDMD inhibitors are clinically effective in the treatment of inflammatory diseases, and the deletion of GSDMD in mouse models of various inflammation-mediated diseases can significantly delay the onset of diseases. Necrotic sulfonamides and disulfiram can suppress the oligomerization of GSDMD-NT fragments through modifying GSDMD, preventing the formation of cell membrane pores and even pyroptosis [35,36]. Another study demonstrated that the α-estrogen receptor acts on GSDMD to restrain hepatocyte pyroptosis, and blocking GSDMD reverses hepatocyte pyroptosis induced by α-estrogen receptor deletion and improves hepatocyte lipid accumulation, metabolic dysfunction, IR and liver damage [37]. GSDME is also composed of three parts and is similar in structure to GSDMD. GSDME was initially considered a “deafness gene” and was later found to be associated with tumor suppression; it participates in pyroptosis downstream of caspase-3 and can convert tumor necrosis factor-α (TNF-α)-mediated apoptosis to pyroptosis, enhance the activity of chemotherapeutic drugs and improve the nephrotoxicity induced by chemotherapeutic drugs [38,39,40]. GSDME can be directly lysed by killer-cell granzyme B, which targets the same site as caspase-3, activates the target cells and induces pyroptosis without caspase-3 [41].

The cysteinyl aspartate-specific proteinase (caspase) family is a set of highly conserved proteases that participate in pyroptosis and apoptosis, but not in other types of programmed cell death [42,43]. In pyroptosis, the caspase family shear GSDMs to produce cell membrane pores. The caspase-mediated pyroptosis pathways can be divided into three categories. The first type is dependent on caspase-1, a proinflammatory protease, which mediates the classical death pathway; the second one is dependent on caspase-4/5/11 that mediates the non-classical pathway; and the third group induces pyroptosis through other pathways, mainly including caspase-3/8, by acting on GSDMD or GSDME. What is noteworthy is that caspase-1 not only activates and cleaves pro-interleukin (IL)-1β and pro-interleukin (IL)-18 to produce mature IL-1β and IL-18, but also cuts the domain linker of GSDMD to expose GSDMD-CT and GSDMD-NT, which induce pyroptosis by assembling pores in the cell membranes [41].

### 2.2. Pathways of Pyroptosis

The pathway of pyroptosis is affected by many factors, which makes it very complex. One of the most studied pathways is the caspase-1-mediated classical pathway. Pyroptosis produces cytoclasis by activating the inflammasome represented by NLRP3, releasing caspase-1 and resulting in the clipping and polymerization of GSDMD which promotes the formation of pores in cytomembranes. NLRP3/NLRP1/NLRC4 or other inflammasomes, as receptors, recognize pathogenic signals when perceiving a stimulus such as PAMPs/DAMPs and associate with to adaptor protein ASC and pro-caspase-1 to constitute a multiprotein complex. Although the non-classical pathway-mediated pyroptosis is less studied than the classical pathway, many reports have also been published on non-classical pyroptosis. Mouse caspase-11 and human caspase-4/5 directly respond to lipopolysaccharide (LPS) secreted by Gram-negative bacilli and cleave GSDMD, resulting in the release of a toxic p30 fragment from the GSDMD-NT fragment, thereby inducing the non-classical cascade. Pannexin-1 and P2X7 channels control the flow of small molecules into and out of cells through the cell membrane and are also involved in non-classical pyroptosis. In LPS-mediated non-classical pyroptosis, LPS induces the caspase-11-dependent pannexin-1 channel cleavage and ATP release, which in turn activate purinergic P2X7 receptors that mediate cytotoxicity. P2X7 or pannexin-1 deletion can abrogate pyroptosis induced by LPS transfection or LPS treatment [44]. Compared with the first two well-known pathways, the GSDME-mediated pyroptosis pathway has been less studied. Also the newly discovered key protein of pyroptosis GSDME primarily relies on caspase-3 to mediate pyroptosis, and this pathway is of relevance especially in cancer. In the field of cancer treatment, chemotherapeutic drugs can mediate pyroptosis through the caspase-3/GSDME pathway, thereby enhancing the effect of chemotherapy; for instance, a Cordyceps militaris extract can induce caspase-3-dependent pyroptosis in A549 cells through caspase-3/PARP and caspase-3/GSDME pathways [45]. It has been reported that pathogenic Yersinia can lead to caspase-8-dependent pyroptosis after entering cells by inhibiting transforming growth factor β-activated kinase 1, activating and cleaving receptor-interacting protein kinase 1 [46] (Figure 1).

## 3. IR and Pyroptosis

### 3.1. IR in the Liver and Adipose Tissue and NLRP3

Many pieces of evidence suggest that pyroptosis is involved in IR in the liver and adipocytes, which is associated with NLRP3 and leads to the damage of insulin target organs. Hyperglycemia caused by diabetes can directly cause pyroptosis, and the NLRP3 inflammasome is closely related to the pathogenesis of type 2 diabetes mellitus (T2DM) [47]. In the early stage of diabetes, diabetes-induced metabolites such as phospholipase C, diacylmercaptoethanol and activated Protein Kinase C directly lead to pyroptosis; with the aggravation of diabetes, changes such as an increase in LPS are accompanied by non-classical pyroptosis; however, when pyroptosis is overactivated, the expression of pyroptosis-related molecules increases, which will stimulate a positive-feedback mechanism promoting the development of diabetes, consisting in the reduction of the number of pancreatic islet β-cells and the enhancement of IR [48].

Studies have found that rats with high-fat diet-induced impaired glucose tolerance presented inflammatory infiltration and pyroptosis in the liver tissue [49]. Hepatic IR is a key manifestation of arsenic-induced T2DM. 10-week-old SD rats administered NaAsO2 by gavage for 3 months showed glucose intolerant, decreased insulin sensitivity, impaired hepatic insulin signaling pathways, the upregulation of oxidative mitochondrial DNA, mitochondrial phagocytosis and the activation of the NLRP3 inflammasome, leading to hepatic IR [50]. It was also found that mtROS scavengers can help reduce mitochondrial damage and inhibit NLRP3 inflammasome activation [50]. Arsenite methyltransferase, an essential enzyme in arsenic metabolism, also facilitates NLRP3 inflammasome activation by N6-methyladenosine modification during arsenic-induced hepatic IR [51]. Giordano et al. reported that adipocytes triggered the NLRP3 inflammasome pathway through a series of reactions in the presence of obesity, bringing about pyroptosis [52]. In the adipose tissue of caspase-1 or NLRP3 knockout mice, inflammation is reduced, and IR is improved even in mice on a high-fat diet [53]. As a cytosolic deacetylase, Sirtuin 2 deacetylation can modify NLRP3 to promote the activation of the NLRP3 inflammasome, and NLRP3 deacetylation can regulate IR; an inhibitor of NLRP3 reduced blood glucose in high-fat diet-induced obese mice, suggesting that the NLRP3 inflammasome is involved in inflammation and IR [54,55]. Chemokine (C-X-C motif) ligand-14 (CXCL14) is a component of the neuronal circuits related to glucose metabolism and feeding behavior, and its expression in the adipose tissue of obese patients is negatively correlated with the expression of genes encoding proinflammatory molecules [56]. By inducing cells to differentiate into mature adipocytes using a chemical cocktail and then culturing them in a glucose-containing medium for 24 h, it was found that CXCL14 caused pyroptosis in adipocytes in the high-glucose environment, and different concentrations of CXCL14 produced different degrees of pyroptosis, suggesting that CXCL14 is likely to be involved in the regulation of adipocyte pyroptosis [57].

Advanced glycation end products (AGEs) are important substances that activate the NLRP3 inflammasome and contribute to pancreas islet injury, a cause of β-cells dysfunction and even death during aging, and they synchronously affect insulin secretion [58]. Mice injected with AGEs for six weeks developed abnormal glucose tolerance and insulin release, impaired β-cell structure, elevated intracellular superoxide anion levels and upregulation of NLRP3, suggesting that AGEs-induced pancreas islet injury may be related to the inflammation caused by NLRP3 [47]. In terms of the mechanism, the accumulated AGEs promote an excessive activation of NLRP3-linked inflammation by generating Reactive Oxygen Species (ROS), a feature of oxidative stress [59].

### 3.2. IR in Skeletal Muscle and NLRP3

Skeletal muscle serves as the main site of postprandial insulin-dependent glucose uptake and depends on a tangled cascade of phosphorylation–dephosphorylation pathways [60]. In skeletal muscle, insulin binds to the insulin receptor (InsR), leading to the phosphorylation of key tyrosine residues, and then phosphorylated InsR causes insulin receptor substrate (IRS)-1 to migrate to the plasma membrane and become phosphorylated [61]. IRS-1 subsequently activates phosphoinositide 3-kinase (PI3K), which induces Akt (also known as protein kinase B or PKB) phosphorylation and activation, thus promoting the translocation of vesicles containing glucose transporter type4 (GLUT4) from cytoplasm to transverse tubules and sarcolemma to ultimately uptake glucose [62]. GLUT4 is a glucose transport protein in fat and muscle cells, and insulin-mediated GLUT4 translocation and skeletal muscle glucose uptake are invariably significantly reduced in the presence of IR [63].

Since skeletal muscle is a target organ of insulin, skeletal muscle IR may be involved in pyroptosis. Luan et al. found that the expression of NLRP3, ASC, caspase-1, GSDMD and IL-1β in flexor digitorum brevis or soleus muscle fibers of male mice on a high-fat diet for 8 weeks was higher than in the corresponding muscle fibers of mice fed the ordinary diet; MCC950, an inhibitor of NLRP3 inflammasome, promoted GLUT4 translocation in isolated fibers of the flexor digitorum brevis in both groups [16]. Dong et al. [64] also found that the expression of nuclear factor-kappa B (NF-κB), NLRP3, caspase-1, IL-1β and IL-18 in the skeletal muscle tissue of rats with impaired glucose tolerance was significantly increased; they then showed that Huanglian Wendan Decoction, a traditional Chinese medicine, could effectively reduce these levels and control obesity, the insulin resistance index and the insulin sensitivity index. Therefore, it is speculated that the mechanism of Huanglian Wendan Decoction relieving insulin IR and then reversing the impaired glucose tolerance process is related to the regulation of the insulin receptor signaling pathway by the NLRP3 inflammasome pathway. Cho et al. [65] discussed the correlation between lipid metabolism, inflammation and skeletal muscle IR in vitro and explored the potential molecular mechanism of skeletal muscle IR in myoblasts. They found that the overexpression of perilipin 2, the most highly expressed lipid droplet-associated protein in skeletal muscle, activated NLRP3, increased the levels of caspase-1 and IL-1β, inhibited insulin-induced glucose uptake and decreased the expression of insulin receptor substrate IRS-1, indicating that IRS-1 is negatively regulated by NLRP3 and IL-1β.

In addition, it was found that ROS participated in hyperglycemia-associated pyroptosis in diabetic rats [66]. In fact, increased ROS production is considered a key feature of NLRP3 inflammasome activation [67]. Growing evidence suggests that ROS production and activation of NF-κB leading to inflammation are associated with mitochondrial damage [68]. Elevated ROS levels in skeletal muscle with IR cause intracellular oxidative stress, which further worsens mitochondrial dysfunction, damages protein, lipid and DNA, and affects the IRS-1/PI3K/Akt/GluT4 signaling pathway, eventually activating a vicious cycle and aggravating IR [69]. TRX-interacting protein (TXNIP), an alpha arrestin protein related to cellular redox reactions, is considered a link between ROS production and NLRP3 inflammasome activation; it affects the intracellular redox state by regulating the activity of the thioredoxin (TRX) redox system, and the generation of ROS leads to the dissociation of TXNIP/TRX and the interaction of TXNIP with NLRP3 [70]. After increasing fructose intake in rats, it was discovered that ROS-induced TXNIP overexpression played an important role in the activation of the NLRP3 inflammasome [71]. It is worth noting that the expression of TXNIP in skeletal muscle is regulated by insulin [72]. Resveratrol could reduce the inflammasome assembly by inhibiting TXNIP [73]. These results suggest that IR in skeletal muscle may lead to an altered regulation of TXNIP expression, but the exact molecular mechanism has not been determined. In addition, hyperinsulinemia will lead to enhanced oxidative stress in the cells, the assembly and recruitment of inflammasome complexes, and the release of inflammatory factors damaging muscle fibers, causing abnormalities in mitochondrial structure; these effects can be reversed by an NLRP3 inhibitor [74] (Figure 2).

### 3.3. IR in Skeletal Muscle and Other Pyroptosis Molecules

Inflammatory NF-κB is a pivotal signal for mouse macrophages to produce mature IL-1β, which promotes a series of immune diseases and inflammatory reactions when incorrectly regulated. NF-κB can bind to the promoter region of caspase-11 and induce caspase-11 expression [75,76]. The NLRP3 inflammasome activates NF-κB to induce cytokine responses in a sterile environment and in a special inflammatory environment, and the activation of NF-κB promotes the upregulation of NLRP3 [77,78]. Meanwhile, NF-κB plays a key role in palmitate-mediated IR in skeletal muscle. As an inflammatory transcription factor, NF-κB mediates palmitate-induced IR in skeletal muscle cells, and overexpression of NF-κB impairs insulin sensitivity and reduces the net insulin-stimulated glucose uptake, GLUT4 translocation, and Akt phosphorylation [79]. However, fibroblast growth factor-21 is able to inhibit the activation of stress kinases and NF-κB, restore palmitate-reduced glucose uptake and prevent palmitate-inhibited Akt phosphorylation to block palmitate-induced IR in human skeletal muscle myotubes [80].

IL-1β and IL-18 are proinflammatory cytokines of the IL-1 family [81]. IL-1β and IL-18 have different roles in diabetes, with IL-1β contributing to type 2 diabetes by attenuating the insulin secretion function of β-cells, and IL-18 being associated with type 1 diabetes [82]. IL-1β, a protein encoded by a gene with a molecular mass of 30,748 Da, plays a fundamental role in the expansion of the inflammatory responses by the regulation of the expression of apolipoprotein-1 and the production of NO [74]. IL-18, a protein encoded by a gene with a molecular mass of 22,326 Da, is involved in the PI3K/Akt signaling pathway, which is related to energy metabolism in skeletal muscle [83]. In pyroptosis, IL-1β and IL-18 are the straightforward activators of the inflammatory cascade, and their pro-inflammatory effects should not be underestimated. In the muscle tissue of patients with dermatomyositis and polymyositis, upregulated glycolysis can activate the NLRP3 inflammasome, leading to muscle cell pyroptosis; interestingly, the pyruvate kinase isozyme M2 in the muscle tissue and IL-1β in the plasma are present at higher levels in patients expressing anti-signal-recognition particle autoantibodies, providing new possible markers for muscle damage [84].

## 4. Skeletal Muscle Exercise Adaptation and Pyroptosis

Skeletal muscle IR is often accompanied by long-term low-density inflammation, and pyroptosis always relates to inflammation; meanwhile, exercise can alleviate IR through an anti-inflammatory effect; the mechanisms of crosstalk between these three aspects are still unclear. The expression of fibroblast growth factor-2, an adipokine, is enhanced in the adipose tissue and during adipocyte differentiation in mice with high-fat-diet-induced obesity and activates the NLRP3 inflammasome, while exercise training can effectively reverse this situation [85]. After 8 weeks of treadmill exercise and adjustment to a normal diet, obese mice gradually lose weight and show a decrease in skeletal muscle IR and inflammasome markers such as NLRP3 and caspase-1 in their adipose tissue [86]. A high-fat diet induced increased LPS production by Gram-negative bacteria in the gut, while patients with obesity and T2DM were shown to lose weight and have increase insulin sensitivity and reduced NLRP3 and IL-1β expression after reducing their energy supply and exercising [82,87].

Fu et al. [88] compared the effects of treadmill exercise (aerobic exercise) and ladder climbing exercise (resistance exercise) on hepatic inflammation and IR induced by diabetes in rats, proving that exercise therapy indeed alleviated liver inflammation and improved metabolic abnormalities. They pointed out differences between the two kinds of exercise, showing that resistance exercise could dramatically mitigate hyperglycemia and skeletal muscle IR in rats compared with aerobic exercise, while aerobic exercise was beneficial in ameliorating abnormal glucose metabolism, alleviating liver tissue morphological anomalies and pathological structure, and inhibiting the activation of the NLRP3 inflammasome and IL-1β. Hypoxia induced ROS generation and upregulates caspase-1, NLRP3, ASC and IL-1β in mice skeletal muscle; the upregulation of nuclear factor erythroid 2-related factor 2, a master regulator of cellular redox, and the reduction of ROS inhibited NLRP3 inflammasome activation when combined with treadmill exercise [89]. Another study showed that endurance exercise and resistance exercise are more beneficial for the treatment of diabetes; in fact, endurance exercise was shown to inhibit the negative regulators of insulin sensitivity such as ROS and inflammatory factors, while resistance training was shown to enhance the synthesis of positive regulators of insulin sensitivity such as galanin and heat shock proteins. In conclusion, both exercise types enhance mitochondrial function, promote glucose uptake and muscle glycogen synthesis [90].

Pyroptosis may be involved in the inflammatory response and muscle injury induced by centrifugal exercise. It has been found that the expression of TLR4, MyD88, NF-κB, NLRP3, TNF-α and IL-1β in the rat solus subjected to a single-session and one-week centrifugation exercise was significantly higher than in the control group, particularly, the injury levels and inflammation of skeletal muscle induced by one-time high-intensity centrifugation exercise was greater than after of one week, suggesting that the activation of NLRP3 and IL-1β in skeletal muscle induced by centrifugation exercise is related to the TLR4/MyD88 signaling pathway [91]. Grass coral coarse polysaccharide blocks the expression of NLRP3, IL-1β and IL-18 and ameliorates the damage and inflammatory infiltration of skeletal muscle caused by centrifugal exercise, participating in the repair of skeletal muscle cells [92].

In addition, pyroptosis appears to participate in the muscular atrophy mechanism. The increase of free fatty acids caused by obesity and the rise of blood glucose caused by diabetes are both closely related to IR, and impairment of the insulin signaling pathway, a consequence of skeletal muscle IR, also leads to muscle atrophy. IR results in the inhibition of insulin or insulin-like growth factor 1 signaling, the downregulation of PI3K/Akt expression, the decrease in protein synthesis and Forkhead Box O1 (a transcription factor that regulates glucose metabolism, fat generation, and bone mass) phosphorylation and the stimulation of protein degradation through the activation of the ubiquitin-proteasome system, thus causing muscle injury and loss in patients with T2DM [93]. After the differentiation and culture of skeletal muscle myotubes and a treatment with LPS, the expression of GSDMD and IL-18 in skeletal muscle myotubes stimulated was obviously higher than in the control group, and this correlated with the degree of myotube atrophy [94]. Inflammation caused by pyroptosis is perhaps a strong stimulator od muscle atrophy, and the changes occurring in glucose metabolism in skeletal muscle after atrophy are unclear. Ren et al. [95] divided mice into a normal group, a tail suspension group (mice subjected to tail suspension and an intraperitoneal injection of dimethyl sulfoxide), a heme chloride group (mice subjected to tail suspension and an intraperitoneal injection of a heme chloride solution and an equal volume of dimethyl sulfoxide to induce heme oxygenase-1 expression), a ZnPP group (mice subjected to tail suspension and an intraperitoneal injection of the same of heme chloride and ZnPP to block heme oxygenase-1 expression). They not merely found not only that the tail suspension group showed an obvious muscle atrophy compared with the heme chloride group and the expression of NLRP3, ASC and caspase-1 in the tibial anterior muscle was increased but also showed that the induction of heme oxygenase-1 could inhibit NLRP3 inflammasome activation and attenuate muscular atrophy. These results indirectly support the possibility that there is a relationship between pyroptosis and skeletal muscle atrophy.

## 5. Summary

The activation and recruitment of the NLRP3 inflammasome and the inflammatory reaction it triggers have been confirmed to be involved in the pathological mechanism of skeletal muscle IR. Therefore, the NLRP3 inflammasome can be considered a signal molecule marker that may predict the release of inflammatory cells and a crucial indicator of the severity of pyroptosis. In addition to supporting the hypothesis that pyroptosis is involved in the mechanism of skeletal muscle IR, we believe that inhibiting the inflammasome activation or blocking pyroptosis pathways can reduce the inflammatory responses and cell damage, relieving skeletal muscle IR. In regard to the anti-inflammatory function of exercise in skeletal muscle, this paper suggests that aerobic exercise and resistance training may improve pyroptosis in skeletal muscle IR and reveals the role of exercise as a non-drug intervention. However, the definite mechanism by which pyroptosis contributes to skeletal muscle IR is still unclear; therefore, several aspects should be explored further. (1) Most relevant studies are based on GSDMD, NLRP3 inflammasome, etc., while GSDME, also a key execution protein of pyroptosis, is less studied, and its involvement in skeletal muscle IR remains indistinct; (2) In addition to IRS-1/PI3K/Akt/GLUT4 signaling pathway, whether there are other target molecules or signaling pathways involves in cell scorch in skeletal muscle IR; (3) In future studies, it will be worth exploring how exercise interventions combined with NLRP3 inhibitors can treat skeletal muscle IR and control the development of skeletal muscle metabolic inflammation. Therefore, in order to clarify the relationship between pyroptosis and IR in skeletal muscle and to provide new research directions to explain skeletal muscle IR and even skeletal muscle metabolic inflammation, in-depth evidence-based research is needed.

## Figures and Tables

**Figure 1 ijms-23-11638-f001:**
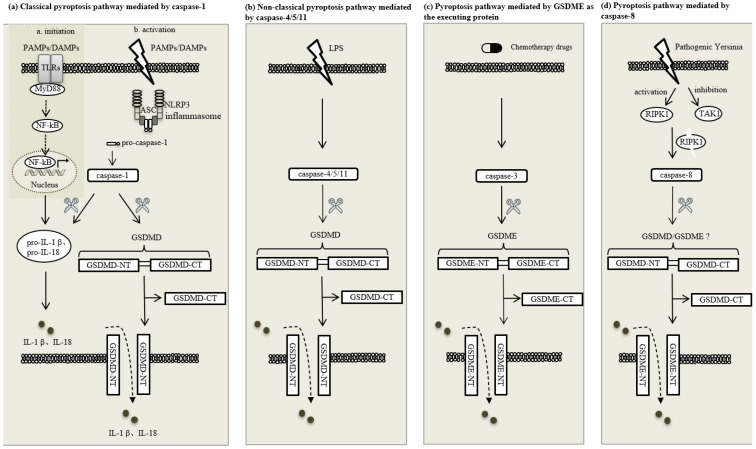
Pathways of pyroptosis. (**a**) Classical pyroptosis pathway mediated by caspase-1; (**b**) nonclassical pyroptosis pathway mediated by caspase-4/5/11; (**c**) pyroptosis pathway mediated by GSDME as the executing protein; (**d**) pyroptosis pathway mediated by caspase-8. The classical pyroptosis pathway involves two steps: initiation and activation. TLRs are Toll-Like Receptors. MyD88 is myeloid differentiation factor 88. NF-κB is nuclear factor-kappa B. PAMPs are pathogen-associated molecular patterns. DAMPs are damage-associated molecular patterns. ASC is speck-like protein containing a CARD. Caspase is cysteinyl aspartate-specific proteinase. GSDMD is Gasdermin D. IL-1β is interleukin-1β. IL-18 is interleukin-18. LPS is lipopolysaccharide. GSDME is Gasdermin E. RIPK1 is receptor-interacting protein kinase 1. TAK1 is transforming growth factor β-activated kinase 1.

**Figure 2 ijms-23-11638-f002:**
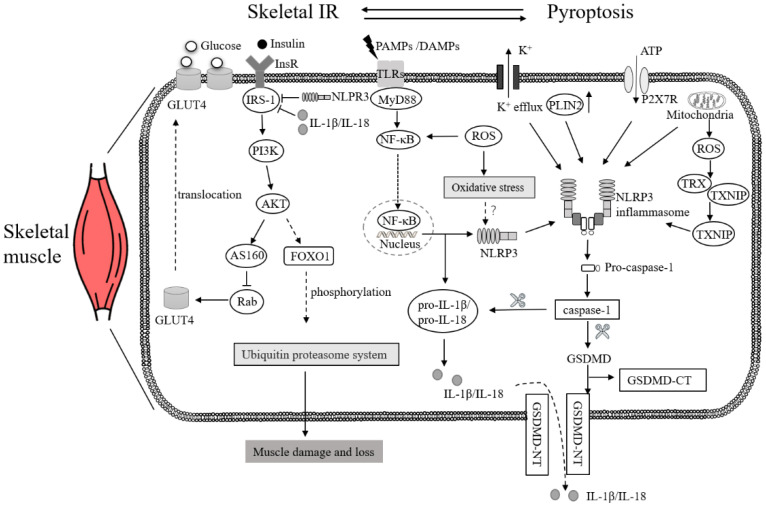
Possible mechanisms of skeletal muscle pyroptosis. GLUT4 is glucose transporter type4. InsR is insulin receptor. IRS-1 is insulin receptor substrate-1. AS160 (is Akt substrate of 160 kDa. FOXO1 is Forkhead Box O1. ROS stands for Reactive Oxygen Species. PAMPs stands for pathogen-associated molecular patterns. DAMPs stand for damage-associated molecular patterns. TLRs stands for Toll-Like Receptors. MyD88 is myeloid differentiation factor 88. NF-κB is nuclear factor-kappa B. NLRP3 is NOD-like receptor pyrin domain-containing protein 3. Caspase is cysteinyl aspartate-specific proteinase. PLIN2 is Perilipin 2. P2X7R is P2X7 receptor. TRX is thioredoxin. TXNIP is TRX-interacting protein. GSDMD is Gasdermin D. IL-1β is interleukin-1β. IL-18 is interleukin-18.

## Data Availability

Not applicable.

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
