# Peer review of "Pyroptosis and Insulin Resistance in Metabolic Organs"

_ijms, 2022, doi:10.3390/ijms231911638_

Round 1
Reviewer 1 Report (Previous Reviewer 4)
The authors describe the emerging mechanisms of pyroptosis in skeletal muscle as well as in liver and adipose tissue. Maybe the title should be adjusted, taking attention on the main focus and the scope the authors would like to reach with the present review. English is really improved, even though some minor spell check is required. The review is full of information and sometime proteins and concepts are introduced without giving to the reader the possibility to follow the flow. I suggest to better re-organize the entire manuscript, making simpler the flow and explaining every new molecular players introduced. Overall, I would have expected to find highlighted changes as always after every resubmission, because it is difficult to understand the changes compared with the previous version of the manuscript. I would strongly suggest to highlight changes if a third submission is planned.
Author Response
Please see the attachment.

Reviewer 2 Report (Previous Reviewer 3)
I have no further comments.
Round 2
Reviewer 1 Report (Previous Reviewer 4)
The authors have addressed the requested adjustments.
This manuscript is a resubmission of an earlier submission. The following is a list of the peer review reports and author responses from that submission.
Round 1
Reviewer 1 Report
1. The relationship among exercise and skeletal muscle insulin resistance, and pyroptosis is poorly demonstrated. The whole section mainly discusses about how exercise improves insulin resistance and inflammation. No direct evidence is provided to show the connection between exercise and pyroptosis.
2. Also, the definition of Pyroptosis should be better discussed. Speculations are just based on uncertain data. Is inflammation always associated with pyroptosis?
3. The main paragraph (section 3) mainly focuses on adipose tissue and liver instead of skeletal muscle; When discussing about skeletal muscle, insulin resistance; inflammation, oxidative stress, and even atrophy are mentioned but not pyroptosis.
4. It has been provided with excessive irrelevant data that cannot support the statements. For example, Line41-44 COVID-19 example;
5. References are lacking. Line p53-56; Line80-83.
6. There are many grammar mistakes and some misspellings
Reviewer 2 Report
The review article titled "Research Progress of Pyroptosis in Insulin Resistance of Skeletal Muscle" addresses an interesting topic concerning the role of pyroptosis in the onset of muscle insulin resistance. Unfortunately the article is written in an incomprehensible way which makes it of poor quality and not publishable.
Reviewer 3 Report
This review tries to push the relatively new hypothesis that pyroptosis may have a role in skeletal muscle insulin resistance and type 2 diabetes. The authors present extensive literature regarding pyroptosis and cell damage gathered from a number of tissues, although the references dealing with this phenomenon in skeletal muscle are relatively scarce. Nevertheless, it is an interesting hypothesis and should be considered. There are numerous grammatical and syntax errors that must be corrected.
Author Response
Thank you for your comment. Please see the attachment.

Reviewer 4 Report
Hard to read and complex organization of sentences that needs rewording and appropriate citations. 74 references could be too few for a normal review.
Here some examples but the review should be completely revised and reorganized:
1) The affiliation of authors is the same, it should be cited with one number and once not twice. Please, correct.
2) Line 32: reference needed. Please reword, sentence too long. Why MS and T2DM are taken together? What is the meaning of the introduction? Please, better explain why you are considering these syndromes and cite valuable references.
3) Line 33: what do authors mean with pathogenesis? Maybe diseases? The pathogenesis is the process that bring to the disease. Not clear.
4) Line 33-35: not clear. Needs English rewording and grammar check. E.g., ‘Growing number of patients’ of what? What do you mean with single treatment? Why single treatment target? Hard to read. It needs also references.
5) Line 35-37: that authors are talking about insulin resistance in general and then they go back talking on T2DM, but the meaning is not clear what they are confuting.
6) Line 41: needs references.
7) Line 41: why the authors cite just data from China? Please, amplify or specify why or at least insert ‘for example’? The IJMS is of world-wide interest.
8) Line44-48: the sentence is complex and needs to be simplified to allow the reader to understand what pyroptosis is in immune system and for other cell types.
9) Line 48: reference n.2 is a comparative study, and the main aim of the paper is not the discovery of pyroptosis but the effect of Pep19-2.5. It is not an appropriate citation. Pleased, change.
10) Line 73: why the authors are talking about what is absent in AIM2 when they are talking about the process of pyroptosis? It seems to be out of the context and makes the reading difficult.
11) Line 82: ‘casepase’ should be correct in ‘caspase’.
12) Line 119: the authors start to use the past since here e.g. acted, was then cleaved, converted. Why? It does not make sense apparently.
13) Line 161: another study has demonstrated not has been demonstrated
Author Response

(The authors gave the same response as above.)
